# Missing *lnc*(RNAs) in Alzheimer’s Disease?

**DOI:** 10.3390/genes13010039

**Published:** 2021-12-23

**Authors:** Rafaela Policarpo, Constantin d’Ydewalle

**Affiliations:** 1VIB-KU Leuven Center for Brain & Disease Research, 3000 Leuven, Belgium; rafaela.policarpo@kuleuven.vib.be; 2Laboratory for the Research of Neurodegenerative Diseases, Department of Neurosciences, Leuven Brain Institute (LBI), KU Leuven, 3000 Leuven, Belgium; 3Neuroscience Discovery, Janssen Research & Development, Janssen Pharmaceutica N.V., 2340 Beerse, Belgium

**Keywords:** Alzheimer’s disease, long non-coding RNAs, gene expression

## Abstract

With the ongoing demographic shift towards increasingly elderly populations, it is estimated that approximately 150 million people will live with Alzheimer’s disease (AD) by 2050. By then, AD will be one of the most burdensome diseases of this and potentially next centuries. Although its exact etiology remains elusive, both environmental and genetic factors play crucial roles in the mechanisms underlying AD neuropathology. Genome-wide association studies (GWAS) identified genetic variants associated with AD susceptibility in more than 40 different genomic loci. Most of these disease-associated variants reside in non-coding regions of the genome. In recent years, it has become clear that functionally active transcripts arise from these non-coding loci. One type of non-coding transcript, referred to as long non-coding RNAs (lncRNAs), gained significant attention due to their multiple roles in neurodevelopment, brain homeostasis, aging, and their dysregulation or dysfunction in neurological diseases including in AD. Here, we will summarize the current knowledge regarding genetic variations, expression profiles, as well as potential functions, diagnostic or therapeutic roles of lncRNAs in AD. We postulate that lncRNAs may represent the missing link in AD pathology and that unraveling their role may open avenues to better AD treatments.

## 1. Alzheimer’s Disease and the Non-Coding Genome: What Is the Link?

Alzheimer’s disease (AD) is a devastating neurodegenerative disorder characterized by a progressive cognitive and functional decline, and the leading cause of dementia worldwide [1,2]. Currently, over 55 million people are expected to live with dementia and despite increasing research efforts over the last years, no disease-modifying treatments are available [1,3]. The extracellular accumulation of amyloid-beta (Aβ) containing plaques and the formation of intracellular neurofibrillary tangles (NFTs) composed of hyperphosphorylated and aggregated forms of Tau protein are well-known neuropathological hallmarks of AD [4,5]. Additional features observed during disease progression include neuroinflammatory responses elicited by microglia and astrocytes as well as neuronal and synaptic loss [4,5].

Most AD cases are classified as sporadic, late-onset AD (LOAD) when there is no (clear) genetic cause and when symptoms usually manifest after the age of 65 years [6]. Conversely, rare monogenic forms of early-onset AD are inherited from autosomal mutations in three genes (*APP*, *PSEN1,* and *PSEN2*) involved in the amyloid-beta (Aβ) precursor protein (APP) pathway culminating in the production and aggregation of toxic Aβ peptides. Nevertheless, twin-based genetic studies of dementia have estimated that LOAD depends on heritability in 60–80% of cases, suggesting that genetics play a crucial role in disease development [7]. The alleles of the *APOE* gene (encoding the apolipoprotein E, APOE), particularly the *APOE* epsilon 4 *(ɛ4)* allele, explain a substantial fraction of this heritability. Still, LOAD etiology is complex and oligogenic, and several genome-wide association studies (GWAS) identified LOAD-associated risk variants in over 40 loci [8,9,10,11,12,13,14,15,16,17,18]. However, many of these genetic variants or single nucleotide polymorphisms (SNPs) are often inherited together by linkage disequilibrium [19]. Additionally, several of these SNPs reside in non-coding regions of the genome which in general still lack functional validation [19,20,21]. Together, these features make it challenging to identify causal genes/variants, regulatory mechanisms, and molecular pathways underlying pathology.

Increasing evidence indicate that a large proportion of the human genome is actively transcribed into non-coding RNAs [22,23,24,25,26,27]. While these non-coding RNAs lack obvious protein-coding potential, they appear to play crucial cellular functions; many of them have been identified as novel regulators of gene expression at the epigenetic, transcriptional, post-transcriptional, and translational levels [28]. Amongst these, long non-coding RNAs (lncRNAs), a subclass of non-coding RNAs typically longer than 200 nucleotides, have been shown to participate in brain development and function, and the dysregulation of their expression implicated in many neurological disorders [29]. In particular, aberrant expression of many lncRNAs has been linked to AD [30,31]. To date, it remains unclear if and how lncRNAs influence AD development and progression.

LOAD-associated SNPs may localize to regulatory DNA elements including for example enhancers and transcription factor binding sites (TFBS). These SNPs may alter gene expression levels that in turn could prompt altered risk to LOAD. The functional effects of SNPs in enhancer DNA or TFBS are not clearly understood and may at least partially also rely on lncRNA-dependent mechanisms. In this review, we will focus on the recent advances and challenges in understanding how genetic variants in lncRNA loci modulate disease susceptibility; we will further discuss how the identification of the molecular pathways, cell type(s), and target genes affected by lncRNAs could pave the way to explore these molecules as potential biomarkers for an accelerated AD diagnosis and subsequent therapeutic intervention at early stages of the disease.

## 2. GWAS and the Identification of Genomic Risk Loci for Alzheimer’s Disease

Over the last years, several GWAS were carried out with the aim to identify the genetic determinants underlying the complex and heterogeneous etiology of LOAD [8,9,10,11,12,13,14,15,16,17,18]. One limitation from GWAS is that often they do not discover causal genes or polymorphisms; instead, they identify regions or haplotypes associated with specific (disease) traits [19]. Furthermore, many variants identified in non-coding regions may confer disease risk by interfering with regulatory elements of the genome, affecting chromatin interactions, and ultimately leading to changes in gene expression levels rather than affecting coding sequences. Hence, genetic mapping and functional characterization approaches remain crucial to identify disease-causative cell types, genes, and variants, and assess their impact on the progression of AD pathogenesis; approaches that are particularly challenging with non-coding variants [19,32].

### Recent Advances in Alzheimer’s Disease GWAS

In 2007, the first two LOAD GWAS confirmed the *APOE* ε4 allele as a major risk factor for AD [33,34]. Since then, several other GWAS and meta-analyses contributed to the exponential increase in the identification of novel AD-associated loci. Just within the last three years, four new AD GWAS were published [9,11,14,18]. In 2018, Marioni et al. reported 27 susceptibility loci [11]. This was followed by a 2019 study with increased sample size, which identified 29 disease-associated loci [14]. In the same year, a GWAS based on clinically diagnosed AD resulted in the identification of 24 risk loci [9]. The latest study by Wightman and colleagues included over a million European individuals as controls; and more than 90,000 samples from either clinically diagnosed cases or people with family history of AD [18]. The authors identified 38 AD-risk loci, including five new loci—*AGRN*, *TNIP1*, *AVCR2*, *NTN5*, *LILRB2*—which had not been associated to any neurodegenerative disorder yet, and two—*TMEM106B* and *GRN*—which were linked to frontotemporal dementia before [35,36,37,38]. Most of the reported variants were also identified in previous GWAS studies [8,9,11,14].

Remarkably, this and other genetic studies point towards a central role for the immune system/inflammatory pathways in LOAD [39]. Functional genomic studies found an association of several coding variants in genes that play a role in microglia and peripheral myeloid cells with AD (e.g., *TREM2* [40,41,42], *MS4A4A/MS4A6E* [43], *ABI3* [42,44], *PLCG2* [42,45], *ABCA7* [46,47], *CD33* [48,49], *PILRA* [50], and *SPI1* [51,52]). However, many of these and other identified genes are involved in multiple AD-associated pathways, such as the amyloid cascade [53,54,55,56,57], Tau pathology [58], lipid metabolism and transport [59], neuronal development and synaptic function [60,61], autophagy [62] or endocytosis [56,63]. Thus, investigating the specific cell type(s), cellular states(s)/pathway(s), and spatiotemporal circumstances in which many of these variants affect disease susceptibility will be crucial to understand the biological mechanisms behind AD pathology.

## 3. A Genetic Link between lncRNAs and Alzheimer’s Disease?

So far, most genetic studies in LOAD focused on the identification of mutations, variants or polymorphisms located near or within protein-coding genes, and how these contribute to the underlying AD pathogenesis [64]. However, top GWAS variants mostly reside in non-coding regulatory regions of the genome outside of protein-coding gene sequences, including promoters, enhancers, and non-coding RNAs [19,20,65,66]. This is not surprising considering that large-scale annotation projects such as GENCODE found over 20,000 non-coding genes in the human genome, of which almost 18,000 consist of lncRNA genes (https://www.gencodegenes.org/human/stats.html accessed on 22 November 2021) [67].

Given the broad range of gene regulatory mechanisms exerted by lncRNAs and increasing evidence demonstrating their involvement in a wide spectrum of brain disorders, it is likely that SNPs located within lncRNA-containing loci might interfere with their biological function and, thus, contribute to AD pathology (Table 1). The AD-associated SNPs rs190982 and rs11771145 are located within two lncRNA genes associated with *MEF2C* (*MEFC2-AS1*) and *EPHA1* (*EPHA1-AS1*) loci, respectively [8]. Recently, the SNP rs3935067 was also identified at the *EPHA1-AS1* locus [18]. The rs2632516 variant was also identified in several AD GWAS and annotated to both a microRNA (*MIR142*) and a lncRNA (*TSPOAP1-AS1*) gene [9,14,17,18]. Another study by Chen and colleagues identified a SNP (rs7990916) located at a brain-specific lncRNA which shows a distinct distribution between cognitively normal elderly, mild cognitive impairment (MCI), and AD subjects [68]. While these SNPs have been associated to LOAD and may confer changes in the expression levels of the lncRNAs they reside in, it is unclear to date how they contribute to AD pathology or AD disease progression.

Based on large-scale GWAS data involved in three ethnicities, microarray data and RNA-seq data analysis, Han et al. identified five lncRNA genes with a potential role in AD [69]. The authors further predicted the function of these lncRNAs using multiple approaches, including genome mapping, expression quantitative trait loci, differential co-expression and gene set enrichment analyses [69]. Four out of the five identified lncRNAs can modify AD susceptibility by regulating genes and pathways involved in the immune system and Aβ-associated mechanisms [69]. In particular, two of these lncRNAs (*NONHSAT018519.2* and *NONHSAT016928.22*), localize within the *BDNF* and *ADAM12* loci, respectively, and are differentially expressed in a region-dependent manner between AD patients and healthy controls within the European ancestry group [69]. Decreased expression of both *BNDF* and *ADAM12* in AD brain samples has been reported, and these genes code for proteins which have been implicated in AD pathology [70,71,72,73].

The occurrence and frequency of specific genetic variants might be reflected on the population ancestry making it essential to replicate GWAS data in different ethnicities [74,75]. A recent study investigated 18 SNPs associated with LOAD in European-based studies in 150 AD patients and 114 controls from the South Brazilian population [76]. Despite the limitation in sample size, four SNPs were found to overlap between both populations [76]. Additionally, eight out of 54 variants found in linkage disequilibrium with the associated SNPs were located withing lncRNA genes; and six were found potentially involved in AD [76]. Of particular interest was the SNP allele rs769449*A, which was found in linkage disequilibrium with the *APOE* ε4 isoform rs429358*C allele and associated with LOAD susceptibility in the study’s cohort. Another SNP, rs769449, localizes to two overlapping lncRNA genes (*NONHSAT179793.1* and *NONHSAT066732.2*). This latter SNP likely affects the secondary structure of *NONHSAT066732.2,* leading to changes in the interaction of this lncRNA with two miRNAs potentially involved in inflammatory responses [76].

Future efforts to identify AD-associated variants located within non-coding regions, and particularly lncRNA genes, may be fundamentally necessary to understand how specific SNPs modulate the expression and function(s) of these regulatory non-coding transcripts and their associated protein-coding genes. This will open the possibility to functionally validate their biological functions, explore their expression dynamics and assess the impact of their dysregulation in AD-related genes and (ultimately) monitor clinical progression.

**Table 1 genes-13-00039-t001:** AD-risk variants identified within lncRNA genomic loci in AD GWAS.

LncRNA ID	Other IDs	Variant ID	SNP Position (GRCh38.p13)	Associated Gene(s)	Reference(s)
*MEF2C-AS1*	-	rs190982	chr5:88927603	*MEF2C*	[8]
*EPHA1-AS1*	-	rs11771145 rs3935067	chr7:143413669 chr7:143104331	*EPHA1*	[8] [18]
*TSPOAP1-AS1*	*BZRAP1-AS1*	rs2632516	chr17:58331728	*MIR142, SUPT4H1*	[9,14,17,18,77]
*NONHSAT160355.1*	-	rs7232 rs12453	chr11:60173126 chr11:60178272	*MS4A6A*^1^, *TCN1*	[69]
*NONHSAT152299.1*	*-*	-	-	*C4A, C4B, TCF4, GRIP1*	[69]
*NONHSAT016928.2*	*Lnc-DHX32-1:1*	-	-	*ADAM12*	[69]
*NONHSAT016928.2*	*BDNF-AS:20*	-	-	*BDNF*	[69]
*NONHSAT021264.2*	*lnc-FAM180B-2:1*	rs71457224 rs10769282	chr11:47602821 chr11:60178272	*MTCH2* ^1^	[76]
*NONHSAT179794.1*	*AC011481.3*	rs10414043 rs7256200	chr19:44912456 chr19:44912678	*APOC1* ^1^ *APOE*	[76]
*NONHSAT066732.2*	*Lnc-ZNF296-6:1*	rs429358	chr19:44908684	*APOE*^1^, *AC011481.3*	[76]
*NONHSAT179793.1*	-	rs429358	chr19:44908684	*APOE*^1^, *AC011481.3*, *lnc-ZNF296-6:1*	[76]
*NONHSAT187478.1*	*HSALNT0039381*	rs4663105	chr2:127133851	*LOC105373605* ^1^	[76]
*NONHSAT182593.1*	*-*	rs744373	chr2:127137039	*-*	[76]
		rs730482	chr2:127136908	*LOC105373605* ^1^	
*TCONS_00021856*	*LINC01080*	rs7990916	chr13:80065389	*-*	[68]

^1^ Gene identified based on SNP information from: https://www.ncbi.nlm.nih.gov/snp/ accessed on 22 November 2021; - unknown/information not available.

## 4. Long Non-Coding RNAs: A Diagnostic Tool for Alzheimer’s Disease?

Classical AD diagnostics rely on the clinical manifestations of the disease supported by brain imaging approaches and blood/cerebrospinal fluid (CSF) biomarker strategies [78] These include the measurement of Aβ_1-42_/Aβ_1-40_ ratio, Tau and phosphorylated Tau peptides in the CSF; amyloid and Tau position emission tomography (PET) as direct imaging biomarkers for Aβ and Tau pathology, respectively; and volumetric magnetic resonance imaging (MRI) of the brain as a surrogate of neurodegeneration [79].

It is, however, well accepted that the neuropathological mechanisms underlying this disease start decades before clinical symptoms are manifested [80]. Thus, there is still the need to establish a biological definition of AD based on biomarkers that reflect such biological alterations at early stages of the disease. LncRNAs may represent an additional and attractive novel class of biomarkers for several reasons. First, they exhibit highly regulated spatiotemporal expression patterns, particularly in the brain and spinal cord [24,29,81]. Second, many lncRNA are differentially expressed in the brain during progression of various neurodegenerative disorders [29,30]. Last but not least, several lncRNAs can be detected not only in tissues and cells, but also in different body fluids such as CSF and blood [82]. Therefore, exploring the expression dynamics of circulating lncRNAs, and evaluating their combination with the already established AD clinical biomarkers, could help to improve early AD diagnosis, to evaluate disease progression, and to monitor treatment efficacy.

### 4.1. Expression Profile of Specific lncRNAs in AD Brain

Several lncRNAs are aberrantly expressed in AD brains compared to healthy controls, suggesting a strong association between altered lncRNA expression and AD pathology (Table 2) [30,31]. Although a direct contribution to AD pathology of many of these lncRNAs remains unknown, multiple studies have shown their implication in AD-related pathways including Aβ and Tau production and/or clearance, autophagy, neuronal proliferation, and apoptosis [31].

Expression of *MAPT-AS1*, a lncRNA associated with the *MAPT* gene, was recently found to inversely correlate with Tau pathology, decreasing with higher Braak stages [83]. Additionally, it has been suggested that *MAPT-AS1* plays a regulatory role on Tau translation by competing with the *MAPT* mRNA internal ribosome entry site for ribosomal binding [83].

Faghihi et al. characterized *BACE1-AS* as a natural antisense transcript (NAT) associated with the *BACE1* gene [84], which encodes the β-secretase BACE1, an enzyme involved in the amyloidogenic pathway and synthesis of Aβ peptides. BACE1 is highly abundant in the brain, and both its expression levels and enzymatic activity are increased in AD brains. *BACE1-AS* plays a critical role in upregulating the levels of Aβ_1-42_ peptides by increasing *BACE1* mRNA and protein levels in SH-SY5Y, HEK293T, and HEK-SW human cell lines and in mice [84]. Additionally, expression levels of this NAT are increased in the brains of AD patients and in amyloid precursor protein (APP) transgenic mice indicating that *BACE1-AS* levels can serve both as a candidate diagnostic marker and therapeutic target for AD [84].

Similarly, the lncRNAs *BC200*, *17A*, *NDM29,* and *51A* can increase Aβ production, and their expression levels were increased in brain tissue from patients with AD [85]. *BC200* facilitates Aβ production by modulating *BACE1* expression levels [86]. In SH-SY5Y cells overexpressing Aβ_1-42_, knockdown of *BC200* led to a significant reduction in BACE1 levels, increased cell viability, and reduced cell apoptosis by directly targeting *BACE1* [87]. Importantly, the increase in *BC200* levels in the neocortex of AD patients is associated with the severity of the disease [86]. The lncRNA *17A* enhances the secretion of Aβ in neuroblastoma cells in response to inflammatory stimuli by regulating the alternative splicing of the GABA B2 receptor and subsequently abolishing its intracellular signaling [88]. APP synthesis is boosted upon upregulation of the lncRNA *NDM29 in vitro* leading to increased secretion of Aβ [89]. Finally, the lncRNA *51A* drives a splicing shift in the canonical variant A of *SORL1* [90], a known risk gene for LOAD [91]. In AD brains, *SORL1* expression is reduced [92] shifting APP processing towards the β-secretase pathway and promoting Aβ peptide formation [93,94]. Therefore, *51A* could play a role in Aβ generation by inhibiting *SOLR1* expression in patients with AD [90]. *EBF3-AS* is another lncRNA whose expression levels is increased in different brain regions of LOAD patients and in the hippocampus of APP/PS1 transgenic mice [95,96]. According to Gu et. al, *EBF3-AS* positively regulates the expression of *EBF3*, a DNA-binding transcription factor, and promotes neuronal apoptosis in a human neuroblastoma cell line [96]. Another study has revealed the presence of an antisense transcript, *LRP1-AS,* at the *LRP1* locus. Importantly, LRP1 is a receptor that has been implicated in multiple AD-associated pathways [53], such as both clearance and production of Aβ peptides [97], internalization of APOE [98] and Tau uptake [99]. While the role of *LRP1-AS* in AD progression is not entirely clear, Yamanaka et al. have identified this lncRNA as a negative regulator of *LRP1* gene expression [100]. In line with this, *LRP1* mRNA expression is reduced in samples from the superior frontal gyrus of AD patients, while *LRP1-AS* levels are increased in this brain region during pathology [100].

**Table 2 genes-13-00039-t002:** Examples of deregulated lncRNAs in brain tissue samples from patients with AD.

LncRNA ID	Trend	Evaluated Tissue/Samples	Proposed Function	Reference(s)
*MAPT-AS1*	↓ in AD	Hippocampus, Parietal cortex, Temporal cortex (http://aging.brain-map.org/ accessed 22 November 2021); Bulk brain tissue (https://doi.org/10.7303/syn3388564 accessed on 22 November 2021)	Inhibits Tau translation by competing for ribosomal RNA pairing with the *MAPT* mRNA internal ribosome entry site (IRES)	[83,101,102]
*BACE1-AS*	↑ in AD	Parietal cortex; Cerebellum; Superior frontal gyrus; Entorhinal cortex; Hippocampus	Under stress conditions, upregulates *BACE1* mRNA and subsequently BACE1 protein expression, leading to the accumulation of Aβ peptides	[84]
*BC200*	↑ in AD	Superior frontal gyrus; Hippocampus	Might facilitate Aβ production by upregulating BACE1 expression levels	[86]
*NDM29*	↑ in AD	Frontal and temporal cortex	Increases APP synthesis, leading to increased secretion of Aβ peptides	[89]
*51A*	↑ in AD	Frontal and temporal cortex	Drives a splicing shift of *SORL1* from the synthesis of the variant A to an alternatively spliced protein form, which leads to an impaired processing of APP and increased Aβ formation	[90]
*EBF3-AS*	↑ in AD	Cerebellum; Superior frontal gyrus; Entorhinal cortex; Hippocampus	Promotes neuron apoptosis in AD, and is involved in regulating the expression of the DNA-binding transcription factor EBF3	[95,96]
*LRP1-AS*	↑ in AD	Superior frontal gyrus	Negatively regulates *Lrp1* expression by binding to Hmgb2 protein and inhibit its activity to enhance Srebp1a-dependent transcription of *Lrp1*	[100]

### 4.2. General Expression Profiles of lncRNAs in the AD Brain

Additional studies have focused on understanding the overall dynamics of lncRNA expression patterns during AD pathology. Zhou and colleagues used re-annotation of microarray datasets to specifically identify AD-associated lncRNAs from *post mortem* tissue samples [103]. The analysis indicated that dozens of lncRNAs are aberrantly expressed in AD patients compared to age-matched controls [103]. Interestingly, two significantly dysregulated lncRNAs identified in this study are involved in protein ubiquitination and lipid homeostasis, suggesting a role for altered lncRNAs in AD-relevant signaling pathways [103]. Moreover, lncRNA expression signatures could be used to discriminate between AD and control tissue samples with comparable sensitivity and specificity to those from protein-coding genes [103]. However, the number of lncRNAs necessary for optimal sample prediction was much lower than that of protein-coding genes, indicating that lncRNAs might be just as relevant as prognostic tools [103].

In another report, Zhou and colleagues performed a comparative analysis in four distinct brain regions––entorhinal cortex, hippocampus, post-central gyrus, and superior frontal gyrus. This study demonstrated that the expression profiles of many lncRNAs are altered both in a region- and age-specific manner in the AD brain [104]. In addition, the authors used machine learning tools to identify a panel of nine lncRNAs that can discriminate between AD and healthy control cases with a diagnostic sensitivity and specificity of 86.3% and 89.5%, respectively, in two independent cohorts [104]. Wu et al. further confirmed the region-specific changes in the expression patterns exhibited by lncRNAs by comparing gene expression data in six distinct brain regions from AD and control patients [105].

A transcriptomic analysis of human cortical samples including 12,892 known lncRNAs and 19,053 protein-coding genes found differentially expressed transcripts from both types of genes in AD cases compared to control individuals [106]. Co-expression network analysis revealed that three lncRNAs––*RP3-522J7*, *MIR3180-2*, and *MIR3180-3*—are frequently co-expressed with relevant AD risk protein-coding genes [106]. For instance, all three lncRNAs are co-expressed with *S100B*, a protein-coding gene which is linked to AD pathology [107].Additionally, in line with previous reports, many of these transcripts are specifically enriched in the brain compared to other body tissues and expressed in a region-dependent manner emphasizing their potential as diagnostic markers for AD [106]. Magistri et al. performed RNA sequencing analysis from LOAD hippocampus samples and identified 31 NATs and 89 long intergenic non-coding RNAs (lincRNAs) as differentially expressed in AD compared to controls [95]. Differential expression of four lncRNAs was further validated by RT-qPCR in different brain regions. Interestingly, the expression of one of the lncRNAs, named *AD-linc1,* is upregulated in LOAD samples, and its expression is induced in a human neuronal *in vitro* model upon exposure to Aβ_42_ [95]. Another transcriptomic profiling study using samples from LOAD cases further reports the aberrant expression of several lncRNAs in the hippocampus of advanced Braak stage patients [108].

Although aging is the main risk factor for developing AD, disease onset is also influenced by gender. For instance, women represent about two thirds of all people diagnosed with AD [109]. Thus, a recent study using microarray datasets and bioinformatics analysis specifically explored how lncRNA expression profiles associate with both age and gender in AD [110]. The authors observed changes in the expression patterns of 16 age-associated and 13-gender associated lncRNAs in the frontal cortex of AD patients compared to healthy controls [110]. Of these, three gender-associated lncRNAs––*RNF144A-AS1*, *LY86-AS1*, and *LINC00639*––negatively correlate with AD Braak stage; and two age-associated lncRNAs––*LINC00672* and *SNHG19*––positively and negatively correlate with Braak stage, respectively [110]. While the underlying mechanisms of most of these lncRNAs are unknown, pathways involved in neurodegenerative disorders, lysosome, synaptic vesicle cycle, axon guidance, and endocytosis pathways are enriched within age- and gender-associated lncRNAs [110].

In summary, the identification of brain region-, age-, and gender-associated lncRNAs and their differential expression patterns in the human AD brain provide potential targets for further investigating their biological functions. Subsequently, it opens the exciting possibility of developing age- and gender-specific diagnosis, prevention, and precision therapeutic options for patients with AD.

### 4.3. Circulating LncRNAs Expression in AD

Most studies investigating the potential to use lncRNAs as diagnostic and prognostic markers focused on different types of cancer and cardiovascular diseases. However, multiple lncRNAs show significantly altered expression patterns in body fluids from AD patients (Table 3) [82]. Thus, exploring free circulating lncRNA transcripts in CSF, extracellular vesicles/exosomes, blood, or plasma can be an interesting non-invasive approach to detect biomarkers for AD and other neurodegenerative disorders.

Levels of the lncRNAs *BACE1-AS* are increased in plasma samples from 88 AD patients compared to 72 controls from a Han Chinese cohort [111]. Of note, Feng and colleagues also evaluated the expression levels of *17A*, *51A,* and *BC200* in plasma samples and found no differences between AD and controls [111]. In addition, the authors found no correlation between the expression levels of these lncRNAs with either a clinical cognitive test called the mini-mental state examination (MMSE) or age, indicating that the diagnostic value of *BACE1-AS* was independent of these parameters [111]. Another study using a smaller cohort reported that *BACE1-AS* levels in plasma are able to discriminate between control, pre-AD, and full-AD individuals, indicating a predictive value for this lncRNA in AD [112]. Contrarily to the previous report, a positive correlation was found between *BACE1-AS* plasma levels and age, while low MMSE scores were associated with higher levels of *BACE1-AS* [112]. Similar data were recently reported by Wang et al. [113]. The authors evaluated the expression of several lncRNAs in plasma-derived exosomes together with image data from the entorhinal cortex and hippocampus of AD patients [113]. Plasma exosomal levels of *BACE1-AS*, but not *51A* nor *BC200*, were increased in AD individuals [113]. These data contrast with a study showing upregulated levels of the lncRNA *51A* in plasma from AD individuals [114]. Furthermore, integrating *BACE1-AS* plasma levels with MRI data from right entorhinal cortex volume and thickness increased specificity (96.15%) and sensitivity (90.91%) as a combinatorial AD biomarker compared to *BACE1-AS* levels alone [113]. However, no associations were found between *BACE1-AS* expression levels and age or MMSE scores in this study [113].

Both CSF and plasma levels of the lncRNA *MALAT1* are downregulated in AD patients compared to control and Parkinson’s disease (PD) individuals [116]. In line with this, *microRNA-125b*, whose expression is regulated by *MALAT1*, is upregulated in AD samples [116]. *MicroRNA-125b* overexpression enhances Aβ production by increasing APP and BACE1 expression in mouse neuroblastoma Neuro2a APPSwe/Δ9 cells [121]. Another study described a role for this microRNA (miRNA) on Tau phosphorylation and neuronal apoptosis [122]. Interestingly, both *MALAT1* and *microRNA-125b* levels in the CSF but not in plasma can predict the decline in MMSE score at years one, two and three in patients with AD [116]. Though most studies demonstrated a role for *MALAT1* in various cancer types [123], *MALAT1* overexpression was recently reported to prevent neuron apoptosis, promote neurite outgrowth, and reduce inflammation in two AD mouse models [115]. Thus, *MALAT1* could not only represent a valuable biomarker but also a potential therapeutic target to help prevent neuronal loss in AD.

The lncRNAs *RP11-462G22.1* and *PCA3* were previously linked to PD [124], and their expression levels are upregulated in CSF-derived exosomes from both AD and PD patients [117]. While these lncRNAs cannot discriminate AD from PD patients, these data further emphasize a general dysregulation of lncRNAs in neurodegeneration.

Differential expression of lncRNAs between AD patients and healthy controls were also investigated in blood samples. For instance, a microarray analysis from peripheral blood mononuclear cells (PBMCs) found 14 upregulated and 20 downregulated lncRNAs in AD samples [125]. Another study compared the expression profile of lncRNAs in PBMCs from AD individuals with two other neurodegenerative disorders, PD and amyotrophic lateral sclerosis, and controls [118]. In AD patients, a total of 23 genes, including 19 protein-coding genes and three lncRNAs––*CH507-513H4.4*, *CH507-513H4.6* and *CH507-513H4.3*––emerged as differentially expressed [118]. While the roles of these transcripts are currently not known, their levels were exclusively altered in AD-derived PBMCs, suggesting a disease-specific expression pattern [118].

The lncRNA activated by TGF-β, named lncRNA-ATB, is also overexpressed in the CSF and serum of patients with AD [119]. This lncRNA was previously associated to multiple pathologies, including multiple types of cancer [126,127,128], keloids [129], and osteoarthritis [130]. Wang and colleagues explored the role of lncRNA-ATB in a cellular model and found that suppressing the levels of this lncRNA might have a protective effect against Aβ-induced neurotoxicity via regulation of miR-200 expression [119]. The miR-200 family has also been implicated in the pathogenesis of AD and explored as a potential biomarker for the disease [131].

Both mRNA and lncRNA transcripts can contain different miRNAs binding sites indicating that these transcripts can regulate each other by competing with shared miRNA-binding sites; these transcripts are so-called competing endogenous RNAs (ceRNAs) [132]. Very recently, Huaying et al. explored a potential deregulation in ceRNA networks in AD by analyzing mRNA, miRNA, and lncRNA gene expression patterns; Nine lncRNAs are associated with AD, PD, and other neurodegenerative disorders [120]. Furthermore, five of the identified lncRNAs––*SNHG14*/*UBE3A-ATS*, *PART1*, *NNT-AS1*, *AC093010.3* and *ARMCX5-GPRASP2*––were evaluated as a potential combinatorial AD biomarker in two distinct AD cohorts [120]. RT-qPCR data revealed a downregulation of the lncRNA *PART1*, and an upregulation of *SNHG14*/*UBE3A-ATS* in serum samples from AD patients compared to healthy controls [120]. While the role of these lncRNAs in AD pathology is unknown, *SNHG14*/*UBE3A-ATS* has been proposed as a therapeutic target for Angelman syndrome due to its role in silencing the expression of *UBE3A* [133].

Taken together, these studies indicate that exploring the biological functions of lncRNAs is crucial to understand their dynamic expression profiles in a pathophysiological context and evaluate their suitability as biomarkers for AD.

## 5. Conclusions and Future Perspectives

Recent advances in AD GWAS revealed several disease-associated variants mapping to non-coding regions of the genome, including within genomic loci harboring lncRNA genes. Despite the generally low abundance of lncRNAs, expression levels of several of them are dysregulated in the course of AD. However, many studies show contrasting data, and often there is no clear overlap between the differentially expressed lncRNAs between controls and AD. These discrepancies likely reflect both methodological and biological variations. In addition, there are still several limitations to the use of lncRNAs in clinical practice for AD and other brain disorders. For instance, detection of some lncRNAs in circulation can be difficult due to their lower levels of expression; in addition, there is still a lack of consensus on which genes are stable and appropriate to be used as reference genes for circulating lncRNAs; finally, many circulating lncRNAs can be found dysregulated in multiple disorders, therefore, lacking diagnostic specificity. We strongly recommend establishing a cross-sectoral working group where basic and translational researchers jointly develop standard operating procedures for the identification and quantification of lncRNAs as potential biomarker for neurological (and other) diseases. We also advocate for the thorough functional characterization of lncRNAs already linked to AD and other neurodegenerative diseases. Comprehensive analyses of lncRNAs dysregulated in AD will undoubtedly provide new insights into disease pathogenesis. We are convinced that new detailed insights in AD-associated lncRNAs may uncover previously unappreciated links between genes and pathways involved in the disease. Ultimately, these lncRNAs may pave the way for the development of novel and innovative biomarker strategies or therapeutic avenues for AD and other neurodegenerative diseases.

## Figures and Tables

**Table 3 genes-13-00039-t003:** Examples of deregulated lncRNAs in peripheral tissue samples from patients with AD.

LncRNA ID	Trend	Evaluated Tissue/Samples	Role in AD?	Reference(s)
*BACE1-AS*	↑ in AD	Plasma; Plasma derived exosomes	Yes (see Table 2)	[111,112,113]
*BC200*	No change in AD	Plasma	Yes (see Table 2)	[111,113]
*51A*	No change in AD; ↑ in AD;	Plasma Plasma derived exosomes	Yes (see Table 2)	[111] [114]
*MALAT1*	↓ in AD	Plasma; CSF	Reported to prevent neuron apoptosis, promote neurite outgrowth, and reduce inflammation in two AD mouse models	[115,116]
*RP11-462G22.1, PCA3*	↑ in AD (and PD)	CSF derived exosomes	Unknown	[117]
*CH507-513H4.4*, *CH507-513H4.6*, *CH507-513H4.3*	↑ in AD	PBMCs	Unknown	[118]
*LncRNA-ATB*	↑ in AD	CSF; Serum	Suppression of this lncRNA might have a protective effect against Aβ-induced neurotoxicity via regulation of miR-200	[119]
*PART1*	↓ in AD	Serum	Unknown	[120]
*UBE3A-ATS*	↑ in AD	Serum	Unknown	[120]

## Data Availability

Not applicable.

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
