# Peer review of "Missing lnc(RNAs) in Alzheimer’s Disease?"

_genes, 2021, doi:10.3390/genes13010039_

Round 1

Reviewer 1 Report

In the current review the authors summarized the discovery of genetic variation associated to GWAS for late onset Alzheimer's disease (LOAD). Because some of these variants are located in non-coding sequences, the authors discuss further how these non-coding loci can modulate disease.

In the first paragraph, the authors remind about the results of GWAS for LOAD and discuss the discovery of some variant located in long non-coding RNA. Then they provide some interesting discussion about the role of certain lncRNA modulating the expression of key AD genes or pathways. More information should be given to understand better how the function of some lncRNAs was investigated in vitro (for example lane 277, 285), at least to mention the cell type in which the study was done. Then they look at the use of lncRNA as biomarkers in AD from plasma, or CSF.

Overall the review gathers a number of studies that are of interest to understand the function of non-coding elements in the field even though they do not consider regulatory element at the DNA level. Thus, the authors may temperate their introduction

Miscellaneous:

  • All paragraphs are numbered 1…
  • Lane 302 : APOE not ApoE
  • Lane 302 be consistent with MAPT (lane 268-273) instead of TAU (here noted Tau, lane302)
  • Lane 303-304: please clarify LRP1 protein or LRP1 gene expression in the two sentences
  • Lane 309 and 321 “and colleagues” or “et al”

Reviewer 2 Report

This manuscript reviewed the potential involvements of long non-coding RNAs in AD. The topic has not been presented to the AD research with much attention, which will draw much attentions and interests to readers.

However, authors presented the review as if long non-coding RNAs are independent from the known and main streams of biomarkers or progressions of the AD pathology. Many of the references authors mentioned in the text and tables included their possible correlations with biomarkers or suggested the pathways and imbalances in the patients. Hence, authors need to present the detailed correlations between biomakers/pathology and long non-coding RNAs, especially causal relationships.

Reviewer 3 Report

This is a nicely written and thorough examination of the rapidly developing field of lncRNA in AD. It also makes clear the difficulty of reconciling the various data into a coherent overview owing to the disparity in methodology. The tables included offer an interesting collection of the data but are not adequately indicated in the manuscript. 
